# MODIFYING MEMORIES IN TRANSFORMER MODELS

## ABSTRACT

Large Transformer models have achieved impressive performance in many natural language tasks. In particular, Transformer based language models have been shown to have great capabilities in encoding factual knowledge in their vast amount of parameters. While the tasks of improving the memorization and generalization of Transformers have been widely studied, it is not well known how to make transformers forget specific old facts and memorize new ones. In this paper, we propose a new task of *explicitly modifying specific factual knowledge in Transformer models while ensuring the model performance does not degrade on the unmodified facts*. This task is useful in many scenarios, such as updating stale knowledge, protecting privacy, and eliminating unintended biases stored in the models. We benchmarked several approaches that provide natural baseline performances on this task. This leads to the discovery of key components of a Transformer model that are especially effective for knowledge modifications. The work also provides insights into the role that different training phases (such as pre-training and fine-tuning) play towards memorization and knowledge modification.

## 1 INTRODUCTION

Large-scale Transformer based language models (Vaswani et al., 2017; Devlin et al., 2018; Radford et al., 2019; Raffel et al., 2019; Brown et al., 2020) have not only pushed state-of-the-art on standard natural language processing (NLP) benchmarks such as GLUE and SQuAD, but they have also been crucial for improving various real-world systems (see, e.g., Nayak, 2019; Rao et al., 2019).

Given that these models are pretrained on a large corpora of text such as Wikipedia and BookCorpus (Zhu et al., 2015), it's quite conceivable that they are able to *implicitly memorize* the factual knowledge in their large number of parameters. Recent works (Petroni et al., 2019; Roberts et al., 2020) have verified this hypothesis by evaluating the pretrained language models on factual knowledge based tasks. This line of work shows that pretrained large Transformer based language models achieve non-trivial performance on various open-domain question answering (QA) tasks that probe the factual knowledge stored in the model parameters.

The aforementioned memorization capability of Transformers opens up many exciting opportunities. In addition to improving generalization with better language understanding, Transformers may also replace or assist traditional knowledge bases (KBs) that are either manually curated or require significant amount of supervision (Roth & Yih, 2002; Kambhatla, 2004; Surdeanu & Ji, 2014). Different from conventional KBs that explicitly memorize factual knowledge, Transformers implicitly memorize knowledge in their model parameters. As a result, Transformers lack one key advantage of the conventional databases: *efficiently modifying* the factual knowledge stored in the model. Unlike Transformers, in conventional databases such as SQL and NoSQL that explicitly store knowledge in the forms of structured tables, key-value pairs, wide columns, graphs, or documents, updating knowledge is straightforward. Knowledge-augmented Transformers, which leverage factual knowledge bases to improve their feature representations, cannot effectively modify their predictions by only updating the symbolic knowledge as it causes conflict with the implicit memorization in their parameters (Verga et al., 2020).

This raises the natural question: Can Transformers cope with the ever-changing world where knowledge is continuously being added, updated, and deprecated? To answer this question, we propose a new task of *explicitly modifying specific factual knowledge in Transformer models while ensuring that model performance does not degrade on the unaltered facts*. This task is useful in many scenarios. For example, the factual knowledge stored by the model can become stale over time, which

needs to be updated periodically, e.g., a sports player may play with different teams over time. Users may ask a Transformer-based assistant model to update certain knowledge (factual or otherwise) that they asked model to memorized in the past, e.g., their favorite tourist destination. In the context of privacy one may need to overwrite unintendedly memorized sensitive information without retraining the model (Carlini et al., 2019). Furthermore, language models are susceptible to various biases present in the large corpora of text used for their training, and such biases may need to be eliminated to ensure a fair application of such models in real-world (Bolukbasi et al., 2016; Bordia & Bowman, 2019; Blodgett et al., 2020).

To the best of our knowledge, this is the first work studying reliable and efficient modification of the factual knowledge memorized by Transformers. The paper makes the following contributions.

- We create a new benchmark to evaluate the ability of a candidate method to modify the factual knowledge of a Transformer model as desired while preserving the model's performance on the unmodified factual knowledge (§ 3.1).

- We formulate the knowledge modification as a constrained optimization problem with a constraint on the loss on the unmodified facts and explore better baseline methods to approximately enforce this constraint (§ 3.3).

- We show that constrained layer-wise fine-tuning is a simple yet effective way to modify the knowledge memorized by Transformers (§ 4).

- We find that it is not necessarily easier to modify factual knowledge in the models that employ explicit memory modules, e.g., FaE (Verga et al., 2020), as compared to those Transformer models that solely rely on implicit memorization.

## 2 RELATED WORKS

Traditionally, KBs are commonly utilized to store and access the relational knowledge in NLP domain (Ji et al., 2020; Zelle & Mooney, 1996; Zettlemoyer & Collins, 2005, inter alia). However, the recent success of Transformer-based language models on a multitude of NLP tasks has fueled an increasing number of efforts on exploring the ability of these language models to serve as unstructured/non-symbolic KBs.

**Language models as a source of factual knowledge.** To assess the performance of off-the-self modern language models as KBs, Petroni et al. (2019) introduced LAMA (LAnguage Model Analysis) probing method that convert various facts and fact-seeking question-answer pairs into cloze sentences. Petroni et al. (2019) concluded that pretrained BERT (Devlin et al., 2018) shows factual knowledge that is competitive with KBs generated using some of the traditional off-the-self techniques. Further, Roberts et al. (2020) probed the knowledge within T5 models (Raffel et al., 2019) and found very promising results. Another line of work (Sun et al., 2019; Zhang et al., 2019; Peters et al., 2019) focuses on leveraging the readily available structured KBs to further complement the knowledge possessed by language models. Earlier works on retrofitting improves word representation learning with relation information (Faruqui et al., 2015). Recently, there have been attempts to develop novel Transformer models and/or training procedures that aim to leverage both available high-quality KBs and large corpora of (unstructured) text (Dhingra et al., 2019; Guu et al., 2020; Lewis et al., 2020), further broadening the scope of factual knowledge. However, unlike structured KBs, which are accompanied by infrastructure for querying, inferring, or updating facts, neural language models do not possess such capabilities directly. Jiang et al. (2020) explored designs for better prompts to query the knowledge implicitly stored in the model parameters of a neural language model. To the best of our knowledge, however, there has been no work on designing efficient ways for modifying knowledge in a neural language model, which is the focus of our present work.

**Memory augmented models.** Multiple recent research efforts augment the Transformer models with explicit long-term memory modules to increase their factual knowledge. Use of knowledge augmented neural networks had been explored in pre-Transformer era as well (Weston et al., 2014; Sukhbaatar et al., 2015). More recently, in the context of Transformers, Févry et al. (2020) utilized an explicit key-value memory to store entity representations, which are trained along with the rest of model in an end-to-end manner. Verga et al. (2020) build on Févry et al. (2020), and introduced Facts as Expert (FaE) model with explicit symbolic memory of $(\mathrm{subject}, \mathrm{relation}, \mathrm{object})$ triples based on end-to-end trained entity representations. Notably, one of the motivations behind FaE is

the ease of updating knowledge by directly modifying the content of the explicit symbolic memory. However, even though FaE has successfully demonstrated injecting new facts to its knowledge base, it exhibits poor performance when one tries to modify the facts that the model encountered during the training due to contradictions between the implicit knowledge of the underlying Transformer model and explicit content of the symbolic memory (Verga et al., 2020, §5.3). Modifying the value tokens in the datastore of kNN-LM (Khandelwal et al., 2020) is another non-parametric method to update the facts. However, this approach tends to cause wrong predictions for all other facts that shared the same object before modification, resulting in low accuracy on the unmodified facts (cf. Appendix F). Thus, our work on modifying the implicit memory of Transformer models also has utility for the task of updating knowledge in memory augmented Transformer models.

**Generalization often requires memorization.** In general, without specifically focusing on language models, Feldman (2020); Feldman & Zhang (2020) have demonstrated both theoretical results and empirical evidences to imply that close-to-optimal generalization requires memorization of labels for samples from the low-frequency sub-populations. This line of work is further supported by the recent efforts on adding the $k$-NN component to language models to improve their generalization via memorization (Kassner & Schütze, 2020; Khandelwal et al., 2020). We believe that our work on modifying the implicit memories in Transformer models can improve their generalization by boosting their factual knowledge in specific domains.

**Memory modification vs. continual learning.** Continual learning, with recent extensions to language models (Sun et al., 2020; Liu et al., 2019; Mi et al., 2020; Chuang et al., 2020), aims to learn a new task while preserving the performance on the previous tasks without access to their data. Similar to continual learning, memory modification also expects the predictions to be updated efficiently (potentially without access to the unmodified facts) while preserving the accuracy for the unmodified facts. In this case, both settings suffer from catastrophic forgetting (Kirkpatrick et al., 2017), but memory modification further requires the model to memorize new facts that conflict with previously learned facts, posing new challenges to existing continual learning approaches, e.g., we may need to update the Gradient Episodic Memory (Lopez-Paz & Ranzato, 2017) or the Conceptors (Liu et al., 2019). Furthermore, our benchmark and the evaluated models are at larger scales as compared to the works mentioned above, posing a stricter requirement on the scalability of the proposed solution.

## 3  MODIFYING IMPLICIT FACTUAL KNOWLEDGE OF TRANSFORMER MODELS

In this section, we define a new knowledge modification task. We then present several approaches to solve this task with different computational costs. We focus on a constrained optimization-based approach that is highly effective and efficient.

### 3.1  MODIFICATION OF IMPLICIT KNOWLEDGE

We propose a new task of modifying specific pieces of knowledge in a model that are stored implicitly in its weights. Specifically, we would like to change the model's weights in a way so that a pre-selected subset of its knowledge is updated, while the rest of its knowledge is preserved. Such modifications can be challenging as each fact is stored non-locally across a large number of weights and each weight can affect a large number of implicitly memorized facts.

More formally, a pretrained Transformer based language model is defined by its parameters $\theta_0 \in \Theta$, which encodes a collection of facts $\mathcal{F}$ that the model has implicitly memorized. We would like to update a desired subset of facts $\mathcal{S} \subset \mathcal{F}$ to a new set of facts $\mathcal{M}$. At the end of the modification process, we should arrive at a model $\theta^{\mathrm{new}}$ that implicitly stores the collection $\mathcal{F}' = \{\mathcal{F} \backslash \mathcal{S}\} \cup \mathcal{M}$. Ideally, the new model $\theta^{\mathrm{new}}$ not only stores the desired modified knowledge, but also retains the performance of $\theta_0$ on the unmodified knowledge $\mathcal{F} \backslash \mathcal{S}$. For example, a Transformer model may have memorized 'Eliud Kipchoge' given the context 'The marathon world record is held by [MASK]'. When another athlete breaks this record, we will need to update this specific piece of knowledge while keeping most of the remaining knowledge intact.

### 3.2  BASELINE APPROACHES

In this subsection we discuss several natural baseline approaches and setup our notation.

**Retraining the model on modified training set.** A natural and reliable approach to solve the afore-mentioned knowledge modification task is to update all the training data, including both the pretraining corpora and the fine-tuning dataset, to be consistent with the new facts, and then fine-tuning the model on the modified training set or even training a new model from scratch to potentially obtain higher success rate. This approach, however, is not practical for modifying a small amount of knowledge: identifying and updating the modified facts in the unstructured datasets is highly non-trivial and retraining the model from scratch is too expensive. Further, the test performance on the modified facts should be approximately the same as the test performance on other facts in expectation, which means we may not achieve high accuracy on the modified facts if the model does not have high overall accuracy in the beginning.

**Fine-tuning on modified facts.** Another natural and efficient approach is to fine-tune the model on the supporting evidences for the modified facts $\mathcal{D}_{\mathcal{M}}$. Such a collection of evidence is not necessarily from the training set; it can be constructed from the modified facts just to change the model's prediction. With $\theta_0$ as the initialization, we solve:

$$\text{minimize}_{\theta \in \Theta} \quad \frac{1}{m} \sum_{x \in \mathcal{D}_{\mathcal{M}}} L(x; \theta), \tag{1}$$

where $m = |\mathcal{D}_{\mathcal{M}}|$ denotes the number of supporting evidences corresponding to the facts to be modified; and $L(x; \theta)$ denotes per-instance loss employed during the fine-tuning process. This approach indeed achieves high accuracy on the modified facts. But due to overfitting and catastrophic forgetting, the model's knowledge about the unmodified facts $\mathcal{F} \backslash \mathcal{S}$ can significantly degrade, as we demonstrate in our experimental studies (cf. § 4.5.1).

**Fine-tuning on a mixture of modified and unmodified batches.** To obtain a higher-than-average accuracy on $\mathcal{M}$ while preserving the accuracy on $\mathcal{F} \backslash \mathcal{S}$, another natural baseline is to use evidences of both $\mathcal{M}$ and $\mathcal{F} \backslash \mathcal{S}$ in every iteration to fine-tune the model. As detailed in Appendix B, this biases the optimization trajectory towards the modified facts. Due to such imbalance, catastrophic forgetting still happens when only using mixed batches in our preliminary experiments. However, when used together with the constrained fine-tuning (cf. § 3.3), this approach could improve the results (cf. Table 4).

## 3.3 CONSTRAINED FINE-TUNING ON SUPPORTING EVIDENCES FOR MODIFIED FACTS

We explore a simpler yet more effective approach for knowledge modification, where we fine-tune the original model only on the modified facts $\mathcal{D}_M$ while using explicit constraints on the weights $\theta$ to achieve minimum interference with the unmodified facts[1]. With the complexity that scales only with the number of modifications, this approach works surprisingly well in memorizing the new knowledge while preserving the unmodified facts.

In the ideal scenario, instead of (1), the model should learn the new facts while keeping the loss small on unmodified facts:

$$\text{minimize}_{\theta \in \Theta} \quad \frac{1}{m} \sum_{x \in \mathcal{D}_{\mathcal{M}}} L(x; \theta) \quad \text{subject to} \quad \frac{1}{n} \sum_{x' \in \mathcal{D}_{\mathcal{F} \backslash \mathcal{S}}} \big( L(x'; \theta) - L(x'; \theta_0) \big) \leq \delta. \tag{2}$$

With a small positive constant $\delta$, we aim to add a constraint on the model's performance on all $n = |\mathcal{D}_{\mathcal{F} \backslash \mathcal{S}}|$ training samples that provide supporting evidences for the unmodified facts $\mathcal{F} \backslash \mathcal{S}$.

However, it is expensive to enforce this constraint. So we approximate the constraint by using local continuity of the loss around $\theta_0$ to obtain the following program:

$$\text{minimize}_{\theta \in \Theta} \quad \frac{1}{m} \sum_{x \in \mathcal{D}_{\mathcal{M}}} L(x; \theta) \quad \text{subject to} \quad \|\theta - \theta_0\| \leq \delta, \tag{3}$$

where $\| \cdot \|$ denotes any suitable norm in the parameter space. We tried $\ell_2$ and $\ell_\infty$ norms in our experiments, where $\ell_\infty$ consistently leads to more stable results for knowledge modification. We

---

[1]We also extend constrained fine-tuning to the mixture of modified and unmodified batches (cf. Appendix B).

solve this problem with projected gradient descent, see Appendix D for details. We also provide a potentially better yet more costly alternative using the Fisher information in Appendix C.

Note that, if we use a very small $\delta$, the model will not change much and the accuracy on the modified facts will be low while the accuracy on the unmodified facts will remain high. If $\delta$ is too large, we are essentially solving (1) which results in almost zero accuracy on the unmodified facts. Therefore, $\delta$ is an important design parameter that needs to be chosen carefully.

**Fine-tuning specific Transformer blocks.** When fine-tuning large models on a small amount of data, a commonly used approach is to fine-tune only a small portion of the model (e.g., one layer) while keeping the rest of the model frozen. Note that, with appropriately chosen $\delta$ to avoid overfitting, full-model fine-tuning and 1-layer fine-tuning will explore very different functional spaces and the later is not contained in the former.

We found that fine-tuning the initial and final Transformer blocks of Transformers results in better adaptation to the modified facts and better preservation of performance on the unmodified facts (cf. § 4). This approach, interestingly, outperforms the case when the whole network is updated. This is partially consistent with Houlsby et al. (2019), who demonstrated that fine-tuning top layers of BERT-Base is the best approach for certain tasks, except that we are also interested in retaining the memorization of the unmodified facts. For more work related to the roles of different layers on QA tasks, see e.g. van Aken et al. (2019); Cao et al. (2020). Here, we found that sometimes initial layers give better results.

## 4 EXPERIMENTS

We now conduct a systematic experimental evaluation of different approaches to modifying the knowledge implicitly stored in the parameters of the Transformer model. Similar to prior works on probing the knowledge of language models (Petroni et al., 2019; Roberts et al., 2020), we rely on factual knowledge-based datasets. From two such datasets, we create two new benchmarks for the knowledge modification tasks (cf. § 4.1). We compare the

| Dataset | # question | # facts |
|---|---|---|
| T-REx (training) | 1,282,567 | 34,039 |
| T-REx (test) | 34,039 | 34,039 |
| zsRE (training) | 197,829 | 147,905 |
| zsRE (test) | 59,527 | 47,156 |

Table 1: Statistics of T-REx and zsRE.

performance of the constrained finetuning approach against several baselines (cf. § 3.2) on models such as BERT (Devlin et al., 2018) and ALBERT (Lan et al., 2019). We also test the FaE model (Verga et al., 2020) modifying its implicit and explicit symbolic memory. A summary of the best results of each model is listed in Table 2.

### 4.1 DATASETS AND BENCHMARKS

We construct the benchmark of modified facts from two datasets, T-REx (Elsahar et al., 2018) and *Zero-shot Relation Extraction* (zsRE) (Levy et al., 2017). Each fact, in the form of (`subject`, `relation`, `object`) triples, is supported by multiple evidences. We modify a relatively small subset of facts by changing their objects and consistently updating all their evidences. For illustration, let's look at an example from the zsRE dataset:

   **Fact:** (Della Pia Glacier, continent, Antarctica)

   **Masked evidence (training):** *What is the continent that Della Pia Glacier is located? [MASK]*

   **Masked evidence (test):** *What continent is Della Pia Glacier found on? [MASK]*

The masked word here is "Antarctica". When we modify this fact, we would consistently replace its object "Antarctica" with a similar entity, e.g. "Asia", which is sampled from all objects that share the same relation, according to their frequency in the training set. Note that the training evidence is phrased differently from the test question, reducing the impact of over-fitting to spurious correlations. Please refer to Appendix A for more details of the benchmark construction process.

### 4.2 PERFORMANCE MEASURE

As the model updates its memory with the modified facts, its memory on the unmodified facts may suffer undesirable changes. For example, finetuning a pretrained model on only modified facts with-

out constraints gives high accuracy on them, but almost zero accuracy on the other facts. Therefore, an ideal metric should take both of these accuracies into account. In this work, we use their average as the performance metric:

$$\bar{\mathfrak{A}} = \left( \mathfrak{A}_{\mathcal{M}} + \mathfrak{A}_{\mathcal{F} \setminus \mathcal{S}} \right) / 2, \tag{4}$$

where $\mathfrak{A}_{\mathcal{M}}$ is the accuracy on the modified facts while $\mathfrak{A}_{\mathcal{F} \setminus \mathcal{S}}$ is the accuracy on the unmodified facts. The trade-off between $\mathfrak{A}_{\mathcal{M}}$ and $\mathfrak{A}_{\mathcal{F} \setminus \mathcal{S}}$ can be strongly affected by certain hyperparameters, such as the constraint $\delta$ (cf. (3)) in the constrained optimization approach. In this cases we select the hyperparameter that optimizes $\bar{\mathfrak{A}}$.

### 4.3 MODEL ARCHITECTURES

We work with three Transformer based language models for our experimental study:

**BERT** (Devlin et al., 2018). We evaluate both the uncased BERT-Base and BERT-Large models without whole word mask training, as released by the official repository[2]. The two models have 12/24 Transformer blocks with 768/1024 hidden dimension and 110M/340M parameters, respectively.

**ALBERT** (Lan et al., 2019). We only evaluate ALBERT-XXLarge model, which is the largest ALBERT model from Lan et al. (2019). It has a total of 235M parameters. The weights are shared in each transformer block, so the only option here is to finetune all its blocks on the modified facts.

**FaE** (Verga et al., 2020). FaE adds symbolic memories to BERT-Base. It inherits the entity memory module from EaE (Févry et al., 2020) and adopts an additional fact memory to enhance the representation of the facts. The EaE part already has 367M parameters, comparable to BERT-Large, so FaE is even larger than BERT-Large.

### 4.4 NOTATIONS AND SETUPS

We start from an. off-the-shelf language model pretrained on a large corpus by default. Afterward, we often finetune our model first on the unmodified T-REx or zsRE. This enables the model to achieve reasonable performance on all the original facts before modification. BERT-Base, BERT-Large, ALBERT-XXLarge, and FaE achieve the accuracy of 50.50%, 51.39%, 47.96%, and 60.38% after this process. We use FT to denote such a finetuned model.

There are two natural ways to train a model to update specific memorized facts. The first approach is to train it only on the modified facts $\mathcal{D}_{\mathcal{M}}$, which we denote by FTM. We can also train it with a mixture of modified facts and unmodified facts, sampled from $\mathcal{D}_{\mathcal{F}'}$ in each minibatch. We denote this setting as FTA, since we have access to all facts.

### 4.5 RESULTS

We now present the results for different approaches and models on our new knowledge modification benchmarks. The best results are summarized in Table 2. A major theme across this section is combating catastrophic forgetting of unmodified facts when we update the model on the modified facts. We compared multiple ways to alleviate this. Finetuning on the modified facts (FTM) with $\ell_\infty$ constraints (cf. (3)) on the model's weights seem to work the better than other natural strategies, such as finetuning on a mixture of modified and unmodified facts (FTA). Furthermore, this strategy works even better when applied only to specific layers of the model rather than the full model. In this section we discuss various aspects of these findings with extensive ablation studies.

#### 4.5.1 FINETUNING ON MODIFIED FACTS WITHOUT CONSTRAINTS

For T-REx benchmark and BERT-Base, Table 3 presents the results for finetuning on only modified facts without any constraints, i.e., we employ (1) which is also equivalent to constrained finetuning (3) with $\delta = \infty$. Note that these results are for a setting where we modify $|\mathcal{M}| = 32$ facts from the T-REx benchmark. We present results for modifying a randomly initialized model (RI+FTM), a pretrained model (FTM), and a finetuned pretrained model (FT+FTM) as defined in § 4.4.

---

[2]https://github.com/google-research/bert.git

| Model | BERT-Base FTM | BERT-Base FTA | BERT-Base FT+FTM | BERT-Base FT+FTA | BERT-Large FT+FTM | ALBERT FT+FTM | FaE FT+FTM |
|---|---|---|---|---|---|---|---|
| Best setting | Block 0 | Block 0 | Block 11 | Block 11 | Block 23 | - | AWT |
| $\mathfrak{A}_{\mathcal{F}\setminus\mathcal{S}}$ (%) | 17.69 | 17.53 | 43.40 | 46.47 | 44.70 | 25.56 | 57.38 |
| $\mathfrak{A}_{\mathcal{M}}$ (%) | 71.25 | 70.00 | 77.84 | 74.31 | 72.80 | 75.42 | 75.00 |
| $\bar{\mathfrak{A}}$ (%) | 47.47 | 43.77 | 60.62 | 60.39 | 58.75 | 50.49 | 66.19 |

Table 2: A summary of the best results when modifying 32 facts with constraints on T-REx for various models. Best setting refers to the best subset of weights to finetune. For BERT models, Block $n$ refers to finetuning only its $n$-th Transformer block (layer). For FaE, AWT refers to weights outside its Transformer part (cf. Table 5). $\mathfrak{A}_{\mathcal{M}}$ is the accuracy on the modified facts, $\mathfrak{A}_{\mathcal{F}\setminus\mathcal{S}}$ is the accuracy on the unmodified facts and $\bar{\mathfrak{A}}$ is their average (cf. (4)). Starting with an $\mathfrak{A}_{\mathcal{F}}$ of 60.38%, the memory-augmented FaE has the best $\bar{\mathfrak{A}}$. However, it does not enjoy a better tradeoff between the gain in $\mathfrak{A}_{\mathcal{M}}$ and the drop in $\mathfrak{A}_{\mathcal{F}\setminus\mathcal{S}}$ compared to the BERT models (cf. § 4.6). The training strategies, FT, FTM and FTA are defined in § 4.4.

| Fine-tuned layer | 0 | | 5 | | 11 | |
|---|---|---|---|---|---|---|
| | $\mathfrak{A}_{\mathcal{M}}$ (%) | $\mathfrak{A}_{\mathcal{F}\setminus\mathcal{S}}$ (%) | $\mathfrak{A}_{\mathcal{M}}$ (%) | $\mathfrak{A}_{\mathcal{F}\setminus\mathcal{S}}$ (%) | $\mathfrak{A}_{\mathcal{M}}$ (%) | $\mathfrak{A}_{\mathcal{F}\setminus\mathcal{S}}$ (%) |
| RI + FTM | 19.38 (2.40) | 0.63 (0.12) | 21.25 (1.05) | 0.33 (0.06) | 20.00 (0.68) | 0.53 (0.09) |
| FTM | 75.00 (3.19) | 0.30 (0.03) | 66.25 (2.40) | 0.83 (0.05) | 67.50 (1.12) | 0.49 (0.03) |
| FT + FTM | 77.50 (2.40) | 0.37 (0.02) | 77.50 (1.37) | 15.09 (1.94) | 82.50 (2.27) | 1.12 (0.25) |

Table 3: Fine-tuning BERT-Base without constraints on the modified supporting evidences $\mathcal{D}_{\mathcal{M}}$ of T-REx. $\mathfrak{A}_{\mathcal{M}}$ is the accuracy on 32 modified facts from the T-REx benchmark and $\mathfrak{A}_{\mathcal{F}\setminus\mathcal{S}}$ is the accuracy on the unmodified facts. The results are averaged over 5 independent runs with standard error in parentheses. RI denotes starting from a randomly initialized model with no pretraining. See § 4.4 for the definition of FT and FTM.

The RI models are not pretrained so they have no language understanding ability to begin with. Thus, with limited training data, they exhibits poor accuracy on both the modified and unmodified facts. In contrast, both FTM and FT + FTM models result in non-trivial accuracy on the modified facts. However, they forget unmodified facts. Before FTM, the pretrained model had an accuracy of 28.85% on all the facts and finetuning on the unmodified dataset (FT) improve it to 50.50%. Unconstrained FTM caused their degradation to $\mathfrak{A}_{\mathcal{F}\setminus\mathcal{S}}$, as reported in Table 3.

Another takeaway from Table 3 is that training different layers in a Transformer leads to different outcomes for the knowledge modification task, which also depends on the state of the original model. In Appendix E, we present additional results on the role of different layers for knowledge modification with different numbers of modified facts.

### 4.5.2 FINETUNING ON MODIFIED FACTS WITH CONSTRAINTS

As observed in § 4.5.1, unconstrained finetuning on the modified facts leads to catastrophic forgetting of the unmodified facts. This happens even when we modify a single layer of BERT-Base. As demonstrated in Figure 1 to 3, using a simple $\ell_\infty$ constraint (cf. (3)) on the model's weights in the modification step (FTM) works surprisingly well in controlling this issue. Recall that we select the constraint strength $\delta$ to maximize the average accuracy (cf. § 4.2).

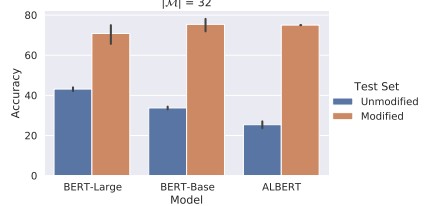

Figure 1: Performance of constrained fine-tuning of all Transformer blocks for BERT-Large, BERT-Base, and ALBERT on T-REx.

These results also demonstrate another interesting effect: the best performances may come from modifying specific layers of the transformer, rather than the entire model[3]. The conclusion comes from combining results from Figure 1 and Figure 2, as well as the results in Figure 3.

Applying a constrained FTM strategy on a single Transformer block ensures good accuracy for both modified and unmodified facts, as long as we modify a small number of facts. However, as the number of modified facts increases, performances degrade, with accuracies on unmodified facts taking

---

[3]This is not possible for ALBERT as it employs parameter sharing across layers.

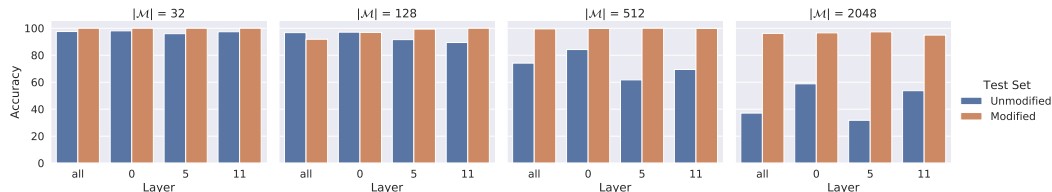

Figure 2: Performance of fact modification for BERT-Base and BERT-Large on the T-REx benchmark. We report the results for the best models obtained by varying $\delta$. The results are averaged over 5 independent runs.

Figure 3: Performance of fact modification for a BERT-Base model on the zsRE benchmark, using the `FT+FTM` setup with constrains during `FTM`. From left to right, the columns show the test accuracy for modifying 32, 128, 512, and 2048 facts, respectively. In each column, we show the best accuracies of constrained finetuning 0th, 5th, 11th, and all Transformer blocks of BERT-Base, which we achieve under different $\ell_\infty$ constraints. The results are averaged over 5 independent runs.

larger hits. In Figure 3, we observe similar results with BERT-Base on the zsRE-based benchmark. We believe this is due to the small model capacity resulting from modifying only one layer.

The best layer for modification also changes with the number of modified facts and the initial state of the model. From Figure 2 and 3, we can see that in the `FT+FTM` setting, as the number of modified facts increases, the block with highest $\overline{\mathfrak{A}}$ changed from the last one (block 11 or 23) to the first one (block 0) for both BERT-Base and BERT-Large. From Table 2, we can see the best block of BERT-Base for modifying 32 facts changed from block 11 to block 0 when starting constrained finetuning from a pretrained model instead of a finetuned model.

### 4.5.3 FINETUNING ON BOTH MODIFIED AND UNMODIFIED FACTS WITH CONSTRAINTS

One obvious reason for forgetting the unmodified facts is that they are excluded from the modification training. Thus, we explore another natural baseline from § 3.2 where we perform constrained finetuning based on a mixture of modified and unmodified facts, i.e., `FTA` in § 4.4. In each minibatch, we use the same number of evidences for modified and unmodified facts. This process implicitly puts more weight on the modified facts since they are usually the minority (cf. Appendix B)[4].

The results for applying `FTA` to different Transformer blocks of BERT-Base on the T-REx benchmark are shown in Table 4. This approach improves the best results, but only by a small margin. Moreover, it performs worse in terms of the weighted accuracy when finetuning 0th or 5th block. These results suggest that when we need to achieve high accuracy on the modified facts, due to the biased optimization trajectory, forgetting some of the unmodified facts might be inevitable even when the model can access them, at least when the weight changes are uniformly constrained.

### 4.6 MODIFYING SYMBOLIC MEMORIES IN A FINETUNED FAE MODEL

An important advantage of the models with symbolic memory modules such as FaE (Verga et al., 2020) is that they could be easily updated by modifying the symbolic links. However, since these

---

[4]Note that if we randomly sample minibatches from $\mathcal{D}_{\mathcal{F}'}$, a finetuned pretrained BERT-Base achieves only $\sim$50% accuracy on the modified facts after training, similar to its accuracy on all facts before modification.

| Fine-tuned layer | 0 | | 5 | | 11 | |
|---|---|---|---|---|---|---|
| | FT+FTA | FT+FTM | FT+FTA | FT+FTM | FT+FTA | FT+FTM |
| $\mathfrak{A}_{\mathcal{M}}$ | 73.31 (0.74) | 72.85 (0.51) | 76.04 (0.65) | 71.09 (0.88) | 70.64 (0.68) | 69.86 (0.46) |
| $\mathfrak{A}_{\mathcal{F} \backslash \mathcal{S}}$ | 18.51 (0.94) | 21.06 (0.31) | 8.73 (0.41) | 16.19 (0.50) | 15.30 (0.50) | 14.71 (0.60) |

Table 4: Comparing the results of finetuning with constraints on the supporting evidence of $|\mathcal{M}| = 512$ modified facts with and without the supporting evidences for the unmodified facts in every mini-batch (T-REx benchmark). We report the results after averaging over 5 independent runs with standard error in parentheses.

| Fine-tuned parameters | NONE | AWT | 3 + AWT | 7 + AWT | All |
|---|---|---|---|---|---|
| $\mathfrak{A}_{\mathcal{M}}$ | 46.88 | 75.00 | 78.12 | 81.25 | 75.00 |
| $\mathfrak{A}_{\mathcal{F} \backslash \mathcal{S}}$ | 60.38 | 57.38 | 45.22 | 41.06 | 53.37 |
| $\Delta\mathfrak{A}_{\mathcal{F} \backslash \mathcal{S}}$ | 0.00 | -3.00 | -15.16 | -19.32 | -7.01 |

Table 5: Results for finetuning different components of a FaE on the $|\mathcal{M}| = 32$ modified facts of T-REx under a range of constraints (FT+FTM setting). $\Delta\mathfrak{A}_{\mathcal{F} \backslash \mathcal{S}}$ is the drop in accuracy on unmodified facts. We report the results with $\mathfrak{A}_{\mathcal{M}}$ closest to the accuracy on the modified facts achieved by the BERT-Large model (77.50%). Surprisingly, FaE does not have a significant advantage in terms of tradeoff between $\mathfrak{A}_{\mathcal{M}}$ and $\mathfrak{A}_{\mathcal{F} \backslash \mathcal{S}}$ when we require $\mathfrak{A}_{\mathcal{M}}$ to be high. AWT (additional weights) refers to all the weights of FaE that are outside its Transformer module, 3 and 7 are the middle and last Transformer blocks of FaE's second-stage Transformer encoder (Verga et al., 2020). NONE refers to finetuning no parameters and modifying only the symbolic knowledge of FaE.

models rely on both the contextual representation and the symbolic links, inconsistency between its implicit memory (realized via contextual representation) and the explicit symbolic memory can result in wrong predictions. In this section, we show that modifying the implicit knowledge is essential for successfully updating these models. We also give results with kNN-LM in Appendix F.

FaE has three key components: a BERT style Transformer model, symbolic memory modules, and model weight connecting the Transformer model with the symbolic memory. We experiment with modifying various combinations of these components as a means to realize knowledge modification (cf. Table 5). Our results show that finetuning the model parameters of FaE in addition to symbolic memory module is necessary for it to obtain high accuracy for the modified facts. Moreover, with constrained finetuning, FAE inevitably experiences a drop in the accuracy for the unmodified facts $\mathcal{F} \backslash \mathcal{S}$, similar to the BERT models *without* explicit memory modules. After modifying the symbolic links stored in its symbolic memory modules, FaE achieves 46.88% accuracy on the modified facts, which is higher than the 30% reported by Verga et al. (2020), and its accuracy on unmodified facts stays unchanged at 60.38%. We find that finetuning only the layers that directly map symbolic memory to the predictions result in the best trade-off (denoted as AWT in Table 5). In particular, after finetuning (AWT), FaE reaches an $\mathfrak{A}_{\mathcal{M}}$ of 75.00% with a drop of 3.00% in $\mathfrak{A}_{\mathcal{F} \backslash \mathcal{S}}$; and an $\mathfrak{A}_{\mathcal{M}}$ of 85.00% with a drop of 6.5% in $\mathfrak{A}_{\mathcal{F} \backslash \mathcal{S}}$ using a slightly larger $\delta$. In contrast, BERT-Large can achieve an $\mathfrak{A}_{\mathcal{M}}$ of 77.50% with a drop of less than 4.00% in $\mathfrak{A}_{\mathcal{F} \backslash \mathcal{S}}$. This indicates that FaE with symbolic memory is not necessarily better than BERT-Large at the knowledge modification task.

## 5 CONCLUSION

We propose a novel task of modifying the factual knowledge implicitly stored in the parameters of a Transformer model. For this task, we introduced two benchmarks based on T-REx and zsRE datasets. We further established the effectiveness of the constrained finetuning approach on the knowledge modification task. We provide comprehensive evaluations for models with and without explicit memory modules, revealing the effect of initial parameters, number of modified facts, and different Transformer blocks on the difficulty of modification. Furthermore, we find that modifying the the Transformer parameters is still necessary for networks with symbolic memory.

While we have explored knowledge modification for models with symbolic fact memory, a more comprehensive exploration of mechanisms to achieve reliable and consistent modification of both implicit and explicit knowledge of such models is an interesting future direction. Another natural future work would be to understand the implications of modifying facts on multi-hop logical inference, i.e. whether the generalization aspect can interact well with modified facts.

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

# Appendix for "Modifying Memories in Transformer Models"

## A    DATASET DETAILS

We aim to construct datasets with a collection of facts $\mathcal{F}$ along with modifications $\mathcal{M}$ for a subset of facts $\mathcal{S} \subset \mathcal{F}$. We take two fact-based datasets, namely T-REx (Elsahar et al., 2018) and *Zero-shot Relation Extraction* (zsRE) (Levy et al., 2017), as the source of the original facts $\mathcal{F}$. These datasets contain a large number of facts (cf. Table 1), with each fact being supported by potentially multiple evidences in the form of *natural-language* masked sentences or cloze-type QA pairs, in which the object of the fact is masked out to serve as a cloze question. This allows a model to memorize a given set of facts by providing such supporting evidences for training. In our experiments, the model learns to predict the masked out object and understands the fact via either memorization of facts from the pretraining datasets (Petroni et al., 2019) or supervised learning on the training sets of T-REx or zsRE. T-REx and zsRE datasets indeed provide different kinds of questions about the same fact. During the test-time, the understanding of a fact by the model is assessed by presenting a cloze-type statement to the model. Note that, it is important to test the model for the given fact using probes that differ from the supporting evidences for the fact in the training set. This is necessary as the model may respond with the correct answer just by overfitting to some spurious correlations present in the pretraining or fine-tuning dataset.

We develop two benchmarks for the knowledge modification task based on T-REx and zsRE. To enable better comparisons with existing works on probing the implicit memorization in language models (Petroni et al., 2019; Roberts et al., 2020), we use the versions of T-REx and zsRE from LAMA (Petroni et al., 2019) and KILT (Petroni et al., 2020) benchmarks, respectively. To modify $m$ facts $\mathcal{S}$ from $\mathcal{F}$, we update the objects in all the cloze-type statements for those facts, which is just the labels of the [MASK] tokens, in both the training and test sets of T-REx and zsRE. The modified object is sampled from the collection of all objects that are connected to the same relation, according to its frequency in the training set. For example, if the original supporting evidence appears in the form of a QA pair, with the question being "*Which country was Charles Darwin born? [MASK]*", we modify the label for the [MASK] token into a random object that appears as someone's birthplace in the training set, other than *United Kingdom*.

**T-REx dataset.** We consider 41 Wikipedia relations with a total of 34039 facts from Petroni et al. (2019). All the object labels in the dataset can be represented by a single token. In this version of the dataset, each fact has at least one supporting sentence (evidence) from Wikipedia with the object replaced by a [MASK] token, plus a template for each relation to construct an additional cloze-type question. We use the masked sentences and the objects from Wikipedia as the training set, and the cloze-type question constructed from the templates as the test set. To enable better comparisons with existing works on probing the implicit memorization in language models (Petroni et al., 2019; Roberts et al., 2020), we use the versions of T-REx and zsRE from LAMA (Petroni et al., 2019) and KILT (Petroni et al., 2020) benchmarks, respectively.

One example of the T-REx dataset:

   **Fact:** (Natalie Lowe, place of birth, Sydney)

   **Masked evidence (training):** *Natalie Lowe (born 15 August 1980), is a professional dancer from [MASK] who has ballroom dancing expertise.*

   **Masked evidence (test):** *Natalie Lowe was born in [MASK].*

For modification, we replace the object *Sydney* with another random object that appears as the birthplace of another subject, e.g., *London*, according to the frequency of the birthplace objects in the training set.

**Zero-shot Relation Extraction (zsRE) dataset.** zsRE is a relation extraction dataset originally formulated as a reading comprehension problem to match each question with a sentence from Wikipedia (Levy et al., 2017). We take the reformulated version of zsRE from KILT (Petroni et al., 2020), which includes multiple template questions for most of the facts. Since the relations in different splits from KILT do not overlap, we construct the modification benchmark from only the training set of zsRE, and split the questions for each fact to obtain the training and test sets for modification. For each fact, we randomly put two of its questions into the test set if it has more than three ques-

tions, preserve the question in the training set if it has only one question, and put one question into the test set otherwise. When applying the uncased BERT tokenizer, we limit the length of the input sequence to be no longer than 512 and the length of the answer to be no longer than 20. We treat a prediction as correct only when all the predicted tokens match the label. One example from zsRE dataset:

**Fact:** (Della Pia Glacier, continent, Antarctica)

**Masked evidence (training):** *What is the continent that Della Pia Glacier is located? [MASK]*

**Masked evidence (test):** *What continent is Della Pia Glacier found on? [MASK]*

## B    FINE-TUNING ON A MIXTURE OF MODIFIED AND UNMODIFIED FACTS

We explore the constrained fine-tuning approach for the knowledge modification task on the T-REx benchmark. Recall that $\mathcal{D}_\mathcal{M}$ and $\mathcal{D}_{\mathcal{F}\setminus\mathcal{S}}$ denote the supporting evidence for the modified facts $\mathcal{M}$ and the unmodified facts $\mathcal{F}\setminus\mathcal{S}$, respectively. The constrained optimization problem becomes

$$\text{minimize}_{\theta\in\Theta} \quad \frac{1}{|\mathcal{D}_\mathcal{M}|}\sum_{x\in\mathcal{D}_\mathcal{M}}L(x;\theta) + \frac{1}{|\mathcal{D}_{\mathcal{F}\setminus\mathcal{S}}|}\sum_{x'\in\mathcal{D}_{\mathcal{F}\setminus\mathcal{S}}}L(x';\theta) \quad \text{subject to} \quad \|\theta-\theta_0\| \leq \delta. \tag{5}$$

Table 4 presents the result for the setting where $|\mathcal{M}| = 512$. We train the model for 10 epochs with a minibatch size of 128, which results in a total of 112 iterations per epoch on $\mathcal{D}_\mathcal{M}$. In each iteration, if using the unmodified training samples, we additionally sample 128 samples from $\mathcal{D}_{\mathcal{F}\setminus\mathcal{S}}$, and compute the gradient of the averaged loss based on the 256 samples. This effectively uses around 10% of the samples of $\mathcal{D}_{\mathcal{F}\setminus\mathcal{S}}$. Such a mixture of modified and unmodified supporting evidence in every iteration is supposed to achieve high accuracy for $\mathcal{M}$, while also preserving the accuracy for $\mathcal{F}\setminus\mathcal{S}$. However, as we observe in Table 4, there is no significant improvement by using such mixed minibatches. Though 50% of the training samples are unmodified evidences in each iteration, the optimizer repeatedly loops over $\mathcal{D}_\mathcal{M}$, which effectively makes the model 10 times as more biased towards minimizing the expected loss on $\mathcal{D}_\mathcal{M}$ (as we train 10 epochs) than on $\mathcal{D}_{\mathcal{F}\setminus\mathcal{S}}$. Such a bias can be alleviated by increasing the ratio of unmodified data in each minibatch, but there would be no guarantee that the model achieves the same level of accuracy on $\mathcal{D}_\mathcal{M}$, even if it is able to improve the accuracy on the unmodified facts.

## C    THE SMALL MODIFICATION LIMIT

In this section we theoretically discuss the small modification limit of the loss constraint in (2), reproduced here:

$$\text{minimize}_{\theta\in\Theta} \quad \frac{1}{m}\sum_{x\in\mathcal{D}_\mathcal{M}}L(x;\theta) \quad \text{subject to} \quad \frac{1}{n}\sum_{x'\in\mathcal{D}_{\mathcal{F}\setminus\mathcal{S}}}\big(L(x';\theta)-L(x';\theta_0)\big) \leq \delta. \tag{6}$$

It is expensive to evaluate the constraint in (6) over the entire $\mathcal{D}_{\mathcal{F}'}$. But in the limit where only a small number of facts are modified and the changes to the weights are small, the constraint simplifies to:

$$\sum_{ij}\Delta\theta_i\Delta\theta_j\frac{1}{2n}\Big(\frac{\partial}{\partial\theta_i}\frac{\partial}{\partial\theta_j}\sum_{x'\in\mathcal{D}_{\mathcal{F}\setminus\mathcal{S}}}L(x';\theta_0)\Big) + \mathcal{O}(\Delta\theta^3) \leq \delta, \tag{7}$$

where $\Delta\theta \equiv \theta - \theta_0$. Here, because the number of modified facts is small, we can assume that we are still at the minimum of the loss function with respect to the unmodified facts. Thus, the linear term in $\Delta\theta$ vanishes and the second order term should dominate.

If we use cross-entropy loss, then the quantity in the bracket (cf. (6)) is the Fisher metric. Even though the Fisher metric only needs to be computed once, it is still expensive as it is difficult to parallelize this computation across samples. We experimented with an approximation of the Fisher information computed with batch size 128, and found that it did not outperform the $\ell_\infty$ norm with (3). We leave the detailed exploration of the Fisher metric for the memory modification task to future work.

# D   SOLVING CONSTRAINED OPTIMIZATION WITH PROJECTED GRADIENT DESCENT

---

**Algorithm 1** Adam with norm constraint

---

1: **Input:** Learning rate $\{\eta_t\}_{t=1}^T$, hyperparameters $0 < \beta_1 < 1, 0 < \beta_2 < 1, \epsilon > 0, \delta > 0$, initial parameter $\theta_0$
2: Set $m_0 = v_0 = 0$
3: **for** $t = 1$ **to** $T$ **do**
4:     Draw samples $S_t$ from training set
5:     Compute $g_t = \frac{1}{|S_t|} \sum_{x_k \in S_t} \nabla L(x_k; \theta_t)$
6:     $m_t = \beta_1 m_{t-1} + (1 - \beta_1) g_t$
7:     $v_t = \beta_2 v_{t-1} + (1 - \beta_2) g_t^2$
8:     $\tilde{\theta}_t = \theta_{t-1} - \eta_t \frac{\sqrt{1-\beta_2^t}}{1-\beta_1^t} \frac{m_t}{\sqrt{v_t} + \epsilon}$
9:     $\theta_t = \Pi_{\|\theta_t - \theta_0\| \le \delta}(\tilde{\theta}_t)$

---

Project gradient descent projects the iterates into the constraint set after each gradient step. In particular, the projection step simply finds the nearest point within the constraint set to the iterate. For the $\ell_2$ norm constraint, the constraint set is $\{\theta : \|\theta - \theta_0\|_2 \le \delta\}$, and the projection operation is

$$\Pi_{\|\theta-\theta_0\|_2 \le \delta}(\theta) = \theta_0 + (\theta - \theta_0) \min \left\{ \frac{\delta}{\|\theta - \theta_0\|_2}, 1 \right\}. \tag{8}$$

For the $\ell_\infty$ norm constraint, the constraint set is $\{\theta : \|\theta - \theta_0\|_\infty \le \delta\}$, and the projection operation is

$$\Pi_{\|\theta-\theta_0\|_\infty \le \delta}(\theta) = \theta_0 + \min \left\{ \max\{\theta - \theta_0, -\delta\}, \delta \right\}, \tag{9}$$

where $\max$ and $\min$ operations are applied element-wise. In our implementation, we use Adam for the gradient step, as shown in Algorithm 1.

# E   ADDITIONAL RESULTS FOR FINE-TUNING WITHOUT CONSTRAINTS

We present additional results for fine-tuning without constraints in Figure 4.

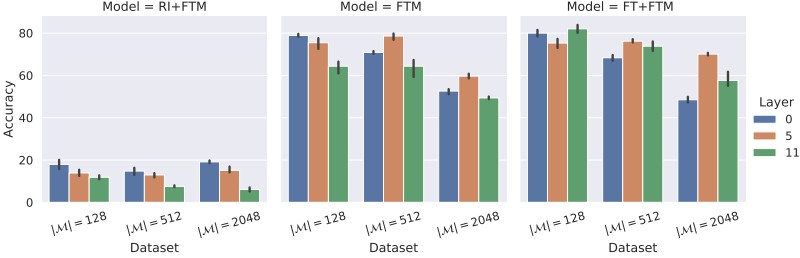

Figure 4: Mean and standard deviation of test accuracies after fine-tuning randomly initialized, pretrained, and fintuned pretrained models on different number of modified facts of T-REx dataset, denoted as `RI+FTM`, `FTM`, and `FT+FTM`, respectively. Here, `RI` refers to starting from a randomly initialized model, "fintuned pretrained model" `FT` refers to starting from a off-the-shelf pretrained model and fine-tune on unmodified T-REx dataset.

# F   kNN-LM FOR MODIFICATION?

kNN-LM (Khandelwal et al., 2020) is originally designed to enhance autoregressive language models with a simple datastore. The datastore is a key-value database, where the keys are the prefix embeddings and the values are the following tokens of the prefixes. During inference, the distribution of the next word is defined as an interpolation between the language model's predictions and a term that decreases with the kNN distances. Without any further training of the language model, kNN-LM improves the results for several language generation datasets.

| $\epsilon$ | 0.5 | 6 | 6.5 | 7 | 8 | 9 | 10 | 11 | 12 |
|---|---|---|---|---|---|---|---|---|---|
| $\mathfrak{A}_{\mathcal{F}\backslash\mathcal{S}}$ (%) | 28.63 | 28.62 | 28.50 | 27.33 | 20.29 | 13.68 | 5.91 | 2.29 | 2.29 |
| $\mathfrak{A}_{\mathcal{M}}$ (%) | 0 | 3.13 | 6.25 | 9.38 | 9.38 | 9.38 | 12.50 | 12.50 | 12.50 |

Table 6: Results for modifying a pretrained BERT-Base model using kNN-LM on $|\mathcal{M}| = 32$ facts from T-REx. $\epsilon$ is defined in Eq. 10, which is the maximum allowable distance for using the nearest neighbor prediction. By comparison, if we modify the 0th Transformer block for the same BERT-Base model, we can obtain $\mathfrak{A}_{\mathcal{F}\backslash\mathcal{S}}$=27.78%/23.51%/17.69% and $\mathfrak{A}_{\mathcal{M}}$=15.63%/58.13%/71.25% with $\delta$ =1e-3/2e-3/4e-3, respectively.

In this paper, we focus on masked language models like BERT. Since we are interested in predicting the [MASK] token, the datastore of the kNN-LM in our setting should be constructed with the keys being the contextual embeddings of the [MASK] tokens from the supporting evidences in the training set, denoted as $c(x; \theta_0)$, and the values being the labels of these [MASK] tokens, which is just the object tokens $y$. The datastore can be constructed on the entire training set, or only constructed for the modified facts to change the model's predictions. Here we focus on the second approach. Specifically, let $f(x; \theta_0)$ be the prediction of the original model (e.g., a pretrained BERT-Base). For a given contextual embedding $c(x; \theta_0)$, we use the prediction from its nearest neighbor in the datastore only when the distance to the nearest neighbor is smaller than $\epsilon$ in the contextual embedding space. Therefore, the model's prediction is defined as

$$f_{nn}(x; \theta_0, \mathcal{M}) = \begin{cases} \arg\min_{\{y'|(z,y')\in\mathcal{D}_\mathcal{M}\}}\|c(x; \theta_0) - c(z; \theta_0)\|_2 & \text{if } d(x; \theta_0, \mathcal{M}) < \epsilon, \\ f(x; \theta_0) & \text{otherwise,} \end{cases} \quad (10)$$

where $d(x; \theta_0, \mathcal{M}) = \min_{(z,y')\in\mathcal{D}_\mathcal{M}}\|c(x; \theta_0) - c(z; \theta_0)\|_2$.

The results are listed in Table 6. We can see that even when we set the $\epsilon$ to a very large value, the model does not have a reasonable accuracy on the modified facts. This indicates that the nearest neighbor does not correspond to the correct fact most of the time, probably caused by the discrepancy between training and test questions regarding the same fact (see the example for the T-REx dataset in Appendix A).

Another fundamental limitation of this approach is that it will potentially modify the answers of all facts sharing the same object if the datastore only contains the modified facts. The masked language model is trained to maximize the score of the prediction on the correct object, achieved by (implicitly) minimizing the distance of the contextual embedding of [MASK] to the embedding of the object's token while maximizing the distance to other tokens through the cross-entropy loss. Therefore, all the contextual embeddings of [MASK] corresponding to the same object should be close if the model makes correct predictions on these samples. If we modify one of the objects, it will conflict with or even lead to wrong predictions on other facts. For example, if we want to modify the birthplace of Charles Darwin from the UK to France, then the kNN-LM will tend to predict France as the birthplace of William Shakespeare as well. Therefore, the tradeoff between the modified and unmodified accuracies is again inevitable in the setting where we only change the values of the datastore of kNN-LM, and it may lead to a worse tradeoff by modifying the predictions on all facts sharing the same object.

If the datastore also contains unmodified facts, during modification, we need to identify all the training samples corresponding to the facts from the unstructured texts, which adds to the difficulty. Even if we can find out all the corresponding training samples, only modifying the values tokens will cause conflict with the datastore of other facts sharing the same object. Thus, we can conclude that finetuning is essential for knowledge modification in the kNN-LM as well.

