# OpenReview forum: "Modifying Memories in Transformer Models"
_ICLR.cc/2021/Conference — Reject_

### Official Review · AnonReviewer4 · 2020-10-27

**Rating:** 5
**Confidence:** 4

**Review:**

The paper proposes a very interesting yet practical problem of modifying the memory encoded in transformer weight to unlearn some facts and update with new facts. This paper seems to be a follow-up work from FAE, which encodes symbolic facts in memory for retrieval. Generally speaking, I like the basic idea of this paper and it might have a broad impact on the whole community. However, there are still a lot of questions about the paper.
1) the paper seems to be written in a rush without refining, there are numerous serious typos and spellings errors, which affect my understanding a lot. For example, in 4.5.1, what is "RT", is it supposed to be "RI"?  In Figure 3, why is the 32, the figures are a mess. Why is the left showing "32->512" while the right showing "32->128"? Why do you say it's sharp degrading, it's not that sharp reflected from Figure 3. I'm not sure if I misunderstand something.
2) The results are also quite messy. The algorithm without constrained optimization has its results reported in the table, while the algorithm with constrained optimization is reported in figures. The results with FAE is yet in another table far away. It's hard for me to compare them and draw a consistent conclusion. Is it possible to aggregate all the main results in one table and demonstrate all the ablation studies using Figures? Currently, the figures involved in 4.5.2 are distributed from page 6 - page 8, is it possible to aggregate them in a concentrated place?
3) Besides these details, I think the proposed method is somewhat "not novel". In lifelong learning or meta-learning community, such constrained optimization algorithms have been explored for a few years to prevent the mode from catastrophic forgetting. I don't think the paper makes any significant contribution to this aspect.
4) Overall, I still quite like the scope of this paper. I would like to see a more structured and clear version of the paper with more fundamental algorithm innovation.

---

> ### Author Response · Authors · 2020-11-19
> **Rebuttal for Reviewer 4**
>
> Thank you for highlighting the importance of the problem we are studying! We have revised the paper to correct the typos. To address your concerns:
>
> *“...in 4.5.1, what is "RT", is it supposed to be "RI"? In Figure 3 .... Why is the left showing "32->512" while the right showing "32->128"? Why do you say it's sharp degrading, it's not that sharp reflected from Figure 3. ”*
>
> 1. Yes, it was our mistake. “RT” should be “RI”.
> 2. Figure 3 (now Figure 2) serves to show the effect of the number of modified facts on the results. For BERT-Base we compare the results of modifying 32 facts with modifying 512 facts. For BERT-Base we compared modifying 32 facts with modifying 128 facts. We will include more consistent comparisons in our future version.
> 3. For example, for BERT-Base, if we modify 32 facts, the accuracy on the unmodified facts is around 45%. If we modify 512 facts, the best accuracy on unmodified facts drops to only 20%, which we think is quite a sharp decline. Similarly, the accuracy on the unmodified facts dropped by around 10% when modifying 128 instead of 32 facts for BERT-Large.
>
> ---
>
> *“Is it possible to aggregate all the main results in one table and demonstrate all the ablation studies using Figures? Currently, the figures involved in 4.5.2 are distributed from page 6 - page 8, is it possible to aggregate them in a concentrated place?”*
>
> We have aggregated the best results of the main models and settings in Table 2 in the current version. We have also rearranged the tables and figures to be closer to their text mentions.
>
> ---
>
> *“Besides these details, I think the proposed method is somewhat "not novel". In lifelong learning or meta-learning community, such constrained optimization algorithms have been explored for a few years to prevent the mode from catastrophic forgetting.”*
>
> The primary goal of our paper is to propose the research problem of how one can modify the facts (implicitly) memorized by Transformer models and construct  benchmarks with reasonable baselines. To the best of our knowledge,  this is the first work studying the reliable and efficient modification of the factual knowledge memorized by Transformers.
>
> One of our interesting findings is that the constrained optimization approach is surprisingly effective, despite its simplicity. There is still a huge room for future works to explore in this newly proposed research area though.
>
> We have added comparisons with lifelong learning to the related works section of our current version. We emphasize that our setting is different from lifelong learning in that it not only requires preserving the performance on the unmodified facts but also requires changing the predictions on the modified facts, causing conflicts with previously learned parameters. Some existing lifelong learning methods may face new challenges in our setting, e.g., we need to update the Gradient Episodic Memory (Lopez-Paz& Ranzato, 2017) or the Conceptors (Liu et al., 2019) on all previous tasks. Computing the Fisher information matrix is also challenging given the size of the models and datasets we evaluated.
>
> We have evaluated not only the constrained optimization approach but also the role of different layers and the impact of using a mixture of modified and unmodified evidences. We also consider knowledge-augmented networks in our setting, which is not a focus of existing lifelong learning methods. This part not only involves the constrained optimization but also the explicit modification of symbolic links of FaE. We provide a more detailed evaluation for knowledge modification than the original FaE paper, involving finetuning different layers of the model to achieve better results, and show that modifying the symbolic link is not enough; we also need to modify implicit memories stored in their parameters to obtain high accuracy on modified facts. Previously, the FaE paper only reported one negative result for modifying the symbolic links on one subset of facts.

---

### Official Review · AnonReviewer1 · 2020-10-28

**Rating:** 5
**Confidence:** 4

**Review:**

The submission proposes and explores the task of modifying factual knowledge in transformer language models. The paper makes a convincing argument that this is a worthwhile problem, as knowledge in existing models quickly becomes out dated, and the cost of re-training from scratch on an updated corpus is prohibitive. The paper suggests several natural alternatives, finding the best results by fine-tuning the model on modified facts, but with a constraint to minimize the difference in from the original model. Surprisingly, the authors further show that it is harder to modify knowledge in the partially symbolic 'Facts as Experts' model than it is in BERT. However, I was unable to follow some important details, and I think the paper is missing an obvious baseline, so I think it needs some more work before it can be accepted.

The proposed method is based around fine-tuning the model  on modified facts, with the hard constraint that the norm of the difference from the original model parameters is less than a threshold. I struggled to find any detail on how the authors enforce these constraints during optimization, and this point should be made clearer.

I'd be really interested to see some more analysis that sheds light on which parameters the transformer is using to store facts. Results in the paper touch on this by exploring fine-tuning different layers, but I think lots of interesting experiments could be added with little extra work. For example, you could try fine-tuning just the word embedding layer, or only the feed forward sub-layers. I think exploring this question would add to the paper, and might improve results.

I also felt the paper was missing an obvious baseline based on kNN-LM (Khandelwal et al. 2019), which is explicitly motivated as a way to add knowledge to transformer language models. For example: first, you could simply encode your the sentence modified facts with the transformer, masking out the modified words. For inference, you could copy a token from the modified facts if it is sufficiently close in representation space, and otherwise predict a token using the baseline transformer model.

More generally, non-parametric methods appear to offer a relatively simple and obvious solution to the task, as facts can be updated by just changing the text. These approaches should be discussed further.


Minor Points
The paper contains frequent grammatical errors (too many to list here), and I'd recommend getting it thoroughly proof read before publication. This did not affect my rating.

---

> ### Author Response · Authors · 2020-11-19
> **Rebuttal for Reviewer 1, part 1**
>
> Thank you for your review and suggestions! We are glad that you find the modification problem worthwhile studying. We agree that having more baselines adds to the strength of the paper, but the non-parametric kNN-LM may not be a strong baseline for the modification task. Please refer to the details below.
>
> *“Surprisingly, the authors further show that it is harder to modify knowledge in the partially symbolic 'Facts as Experts' model than it is in BERT.”*
>
> To clarify, by “We find that it is not easier to modify the memorized facts of memory networks like FaE”, we were trying to express that modifying FaE to obtain the same level of accuracy on the modified facts (around 78% in the setting of modifying 32 facts) requires almost the same or even more drop in the accuracy on the unmodified facts than the BERT-Large model in the experiments we have tried. FaE has a 10% higher test accuracy from the beginning, so the overall (weighted) accuracy of FaE is still the best among the models.
>
> ---
>
> *“I struggled to find any detail on how the authors enforce these constraints during optimization, and this point should be made clearer.”*
>
> We have included the details for solving the constrained optimization with norm constraints on the weights in Appendix D in the revised version.
>
> ---
>
> *“I'd be really interested to see some more analysis that sheds light on which parameters the transformer is using to store facts...For example, you could try fine-tuning just the word embedding layer, or only the feed forward sub-layers.”*
>
> Our current result does shed light on which parameter the Transformer is using to “store” facts, indicated by the effectiveness of each Transformer block for modifying the memory. The landscape is more complicated than we have imagined. We find that it depends on the state of the model, the number of facts to modify, and the finetuning approach. First, the FT process shifts the most effective layer for modification from Block 0 to Block 11 for the BERT-Base model when modifying 32 facts on T-REx (Table 2 and 3 in the current version). Second, as the number of modified facts increases, the most effective layer also changes. In Figure 4 of the current version, we find that for FTM and FT+FTM without constraints, when we modify more than 512 facts, Block 5 turns out to be the most effective one. In Figure 2 and 3, we find that, with constraints, using 512 facts shifts the most effective block from Block 11 back to Block 0.
> We had experimented with fine-tuning only the feedforward layers instead of finetuning whole Transformer blocks, and the results were not as good as the one presented in the paper. We will post results with finetuning the embedding layers when the results are available.

---

> > ### Author Response · Authors · 2020-11-19
> > **Rebuttal for Reviewer 1, part 2**
> >
> > *“the paper was missing an obvious baseline based on kNN-LM (Khandelwal et al. 2019)...non-parametric methods...should be discussed further”*
> >
> > We already have results for nonparametric methods, where we only modify the symbolic links of FaE according to the modified facts (Table 5 “None” in the current version). FaE is a strong baseline among the memory-augmented Transformer models, and the results of updating stale knowledge in their paper enables direct comparisons with our paper.
> > Based on your suggestion, we have added experimental results (using the method you described) and additional discussion about kNN-LM into Appendix F of our current version. From the experiments and analysis, we conclude that modifying the value tokens of kNN-LM alone cannot achieve good results, and finetuning is still necessary for kNN-LM. For your convenience, we post the results for modifying 32 facts from T-REx for a pretrained BERT-Base again here:
> >
> > | Max Dist.                             |  0.5  |   6   |  6.5  |   7   |   8   |   9   |   10  | 11    | 12    |
> > |---------------------------------------|:-----:|:-----:|:-----:|:-----:|:-----:|:-----:|:-----:|-------|-------|
> > | $\mathfrak{A}_{\mathcal{F}\setminus \mathcal{S}}$ (%) | 28.63 | 28.62 | 28.50 | 27.33 | 20.29 | 13.68 |  5.91 | 2.29  | 2.29  |
> > | $\mathfrak{A}_{\mathcal{M}}$ (%)              |   0   |  3.13 |  6.25 |  9.38 |  9.38 |  9.38 | 12.50 | 12.50 | 12.50 |
> >
> > “Max Dist.” refers to the maximum acceptable distance for the nearest neighbor, i.e., we only use the nearest neighbor for prediction when its distance to the cloze question is closer than “Max Dist.” in the embedding space. Please refer to Appendix F for more details.
> >
> > The main cause for the bad performance here is the mismatch between the contextual embeddings of the same fact on the training and test set. By comparison, constrained finetuning of a pretrained BERT model on modified facts achieves unmodified accuracy of 27.78%/23.51%/17.69\% and modified accuracy 15.63\%/58.13\%/71.25\% respectively when setting $\delta$=1e-3/2e-3/4e-3. The finetuned FaE model achieves 46.88% modified accuracy without a drop in unmodified accuracy immediately after modifying its symbolic links.
> >
> > The optimization method proposed in this paper is geared towards modifying the facts implicitly stored in parametric models, rather than modifying the database of a non-parametric memory unit. We recognize these two approaches as complementary research directions. As we have already demonstrated in FaE (Section 4.6), even in memory-augmented models it is essential to also modify its implicit memory via constrained finetuning to achieve good performance.
> >
> > Except for the mismatch between training and test contextual embeddings, kNN-LM, which does not explicitly consider the subject and relation of the fact, may have the following two disadvantages for the modification task: 1) It is difficult to identify the modified facts and update its unstructured datastore, if the datastore is a mixture of modified and unmodified facts. 2) Modifying one fact interferes with predictions on other facts sharing the same object. For example, if we want to modify the birthplace of Charles Darwin from the UK to France by only modifying the value token of the corresponding contextual embedding into France, then the kNN-LM will tend to predict the birthplace of William Shakespeare into France as well, which is undesirable. Therefore, the tradeoff between modified and unmodified accuracies is inevitable, and it may lead to a worse tradeoff by affecting the predictions on all facts sharing the same object simultaneously. Please refer to Appendix F for further discussions.

---

### Official Review · AnonReviewer2 · 2020-10-29
**Timely paper that is of interest to the community, with new benchmark data and baselines; some settings are unconvincing and take-aways are not new**

**Rating:** 6
**Confidence:** 4

**Review:**

This paper studies a new problem: evaluating the ability of modifying knowledge inside the Transformer models when models memorize world knowledge inside its parameters. The contribution of this paper is two-fold: (1) introducing a new benchmark for evaluating such ability, and (2) evaluating a comprehensive list of baselines, including a new model that has a constraint term in the objective during fine-tuning.

Strengths of the paper:

1. The problem is well-motivated and of interest to the wider community. Examining the behavior of transformers given updated facts is a timely topic, given recent progress and interest in large pretrained models storing world knowledge and achieving reasonable performance on downstream tasks (slot infilling, question answering) without access to external knowledge sources.
2. The benchmark dataset is created in a reasonable way - it is created on top of T-REx and Zero-shot RE dataset, where a target entity is replaced by another entity that shares the same relation with the other entity.
3. The constrained objective provides a simple yet nice way of updating knowledge in the model parameters in a constrained manner. This model shows that the fine-tuned model overfits less to the modified knowledge, compared to naive baselines.


Weaknesses of the paper:

1. I am not fully convinced by the setup in the paper, where the model is pretrained on unmodified facts and then fine-tuned on a modified knowledge. First, in a natural setting, a set of knowledge sources will always contain both unmodified facts and modified facts together. Therefore, an assumption in the paper that the model can only access modified facts during fine-tuning seems to be unrealistic. Second, if the research question here is the generalization ability of the model, isn’t a zero-shot setting or a setting with small training examples (e.g. 1k) more suitable?
2. Although the creation process of the benchmark dataset makes sense, it is still created synthetically. Furthermore, the paper does not include data analysis or human performance estimation, making it hard to estimate the quality of the data. This is important because, when a subset of knowledge was synthetically updated, some knowledge will contradict each other.
3. Although the model with constrained objective overfits less to the modified knowledge, it still overfits a lot, achieving significantly lower numbers on unmodified facts. It is still a nice baseline, but the claim in the paper: “best way to enforce the constraint in Transformer models” (Sec 1) seems to be overclaiming.
4. Although the paper includes comprehensive experiments, there isn't really new take-aways that are not different from naive expectations. The conclusion, "the model overfits to modified facts and suffers from catastrophic forgetting" is pretty naive and has been observed in a lot of prior work ([1] is one of recent ones).


Questions

1. Is there a specific reason that the scope of this paper is restricted to transformers? Looks like the general idea can be applied to any model that does not have an access to external knowledge source?
2. The creation process of the benchmark data is strictly limited to structured KBs. Is there a way to create such a dataset for tasks based on unstructured text?
3. Are there baseline numbers for PT+FT (without FTM) reported?

[1] Rolnick et al. Experience replay for continual learning. NeurIPS 2019.

---

> ### Author Response · Authors · 2020-11-19
> **Rebuttal for Reviewer 2, part 1**
>
> Thank you for acknowledging the contributions of our paper! Below we address your concerns.
>
> *“I am not fully convinced by the setup in the paper, where the model is pretrained on unmodified facts and then fine-tuned on a modified knowledge. First, in a natural setting, a set of knowledge sources will always contain both unmodified facts and modified facts together. Therefore, an assumption in the paper that the model can only access modified facts during fine-tuning seems to be unrealistic. Second, if the research question here is the generalization ability of the model, isn’t a zero-shot setting or a setting with small training examples (e.g. 1k) more suitable”*
>
> We have definitely tried to provide the model with both modified facts and unmodified facts in the modification process, see section 4.5.3 and Table 4 (originally in the appendix). Surprisingly, this did not improve the model’s performance on the averaged accuracy, compared to providing the model with only modified facts and using constrained optimization. Only using the modified facts in the modification process is a more efficient approach and it is not difficult to just pick the supporting evidences of modified facts. Also, to clarify, in the FT process, we finetuned the model on the whole original training set of T-REx/zsRE, not just the facts that will not be modified.
> The “generalization ability” in this work refers to the ability to generalize to different paraphrases of the same question, e.g., the model generalizes well if it gives the same answers to “What is the continent that Della Pia Glacier is located?” in the training set and “What continent is Della Pia Glacier found on?” in the test set. Therefore, we construct datasets so that the training set covers all the facts in the test sets, but their questions are asked in different ways.
> It is almost impossible for a model without memory modules to answer factual questions if the model was never trained on samples entailing the fact and implicitly memorized the fact, so the traditional zero-shot setting does not seem practical in our case. We are already operating in the small-training-samples regime as you suggested. During the modification process, we indeed use only a small set (up to thousands) of training samples for modified facts.
>
> ---
>
> *“...the paper does not include data analysis or human performance estimation, making it hard to estimate the quality of the data. This is important because, when a subset of knowledge was synthetically updated, some knowledge will contradict each other.”*
>
> Thanks for pointing out the need for human performance evaluation. However, we believe our current setting does not introduce conflicts with other facts, indicated by the results with FaE, where we see no drop in accuracy on unmodified facts after we modify its symbolic links for the modified facts (Table 5, “None” in current version; also included in the previous version). This is probably because we only modify the object of the facts and the current datasets does not contain multiple possible objects for each subject-relation pair.
>
> ---
>
> *“...best way to enforce the constraint in Transformer models” (Sec 1) seems to be overclaiming.”*
>
> We have rephrased the sentence into “We formulate the knowledge modification as a constrained optimization problem with a constraint on the loss on the unmodified facts and explore better baseline methods to approximately enforce this constraint.” We would like to emphasize that the simple norm constraint is not only efficient but also surprisingly effective compared to other methods we have tried.
>
> ---
>
> *“the model overfits to modified facts and suffers from catastrophic forgetting" is pretty naive and has been observed in a lot of prior work ([1] is one of recent ones)”*
>
> We agree the phenomenon does look similar to prior works on continual learning, but the modification task is different from continual learning in that the modified facts conflict with the original training data, which means the model cannot preserve its behavior on all the previously learned facts. It is a new challenge to existing continual learning algorithms. Please refer to the last paragraph of the related works section (Section 2) in the new version for a detailed  discussion.

---

> > ### Author Response · Authors · 2020-11-19
> > **Rebuttal for Reviewer 2, part 2**
> >
> > *“Is there a specific reason that the scope of this paper is restricted to transformers? Looks like the general idea can be applied to any model that does not have an access to external knowledge source?”*
> >
> > We pick the Transformer models only because it has been quite prevalent and powerful for question answering tasks recently. The same approach does apply to LSTMs and any other networks. We leave such explorations as future work. Also, our method applies to the models with access to external knowledge sources as long as such models have trainable parameters. For example we have already included results for FaE, which is a Transformer model with external knowledge source (cf. Section 4.6), and found constrained finetuning is necessary for FaE to obtain higher accuracies on the modified facts than just modifying its symbolic links.
> >
> > ---
> >
> > *“The creation process of the benchmark data is strictly limited to structured KBs. Is there a way to create such a dataset for tasks based on unstructured text?”*
> >
> > Note the data in the benchmark is still presented in unstructured text. See Appendix A for examples. Even without access to the corresponding supporting evidences as in our current setting, we can still create unstructured text as the training sample for modified facts, and apply our method to modify the predictions on the facts we want to modify.
> >
> > ---
> >
> > *“Are there baseline numbers for PT+FT (without FTM) reported?”*
> >
> > Before finetuning, the BERT-Base model has an accuracy of 50.50% on T-REx. The numbers are 51.39%, 47.96%, and 60.38% for BERT-Large, ALBERT, and FaE, respectively.

---

### Official Review · AnonReviewer3 · 2020-11-02
**Nice idea but bad execution or badly written**

**Rating:** 6
**Confidence:** 4

**Review:**

Summary

Recently, pretrained Transformer language models have been shown to capture world knowledge (using testbeds containing facts). What if you want to update a fact, for example, with the current president of USA? This paper investigates different approaches to update the weights of a Transformer model such that the model works for the modified facts but does not catastrophically forget unmodified facts. The main proposal is a simple regularization technique (which they call constrained fine-tuning) to minimize weight changes while fine-tuning on the supporting factual sentences that represent the modified facts.

Strengths

1. The problem of updating world knowledge in Transformers in itself is an interesting problem and novel.
2. The problem is well motivated and the first half of the paper is well-written.
3. The proposed method of fine-tuning along with regularization is simple and it seems to work better than just fine-tuning methods.


Weaknesses:
1. Confusing experimental section, and many important details are missing (see comments).
2. The paper felt like it is a last-minute submission and written in haste.
3. Important related work on retrofitting literature not cited.
4. The proposed method works at the cost of forgetting unmodified facts as the number of unmodified facts increase.


Comments:


Although the reviewer likes the problem formulation and the main idea, they find it hard to follow the experimental section.

1. There were no details on how to find supporting sentences of target facts that one wants to change. This is a non-trivial task and without a detailed description, the paper is impossible to replicate. There were no examples anywhere including appendix.
2. Why do the authors start with pretrained + fine-tuned model? Isn't pretrained model trained on all sentences anyways? Do they mean separate fine-tuned data that contain factual statements? Why does one need this data? Isn't this against the spirit of pretrained models as knowledge bases?
3. Results of vanilla pretrained models on unmodified and modified facts missing, i.e., PT alone. Also, PT + FT.
4. The notation and acronyms are confusing. The authors use a lot of acronyms like PT, FT, FTA, FTM, RT which is unnecessary. This makes the results table unreadable without reading the paper.
5. No citations on retrofitting literature which is very similar idea to this work (e.g., Faruqui et al. 2015). This limits the novelty of the work but the reviewer give credits to the problem formulation.
6. How is PT+FT+FTM different from PT + D_F'?
7. FTA is introduced but never used.
8. The reviewer had a hard time following experiments with FAE section. What is the main takeaway from these experiments? The model has to be described in detail (perhaps with a figure).
9. Examples, Examples, Examples. Show some examples from each dataset.

---

> ### Author Response · Authors · 2020-11-19
> **Rebuttal for Reviewer 3, part 1**
>
> Thank you for your valuable feedback. We are glad that you find this task novel and interesting. We have revised the paper and added discussions about retrofitting literature into the related works. We believe the readability of this work is significantly improved in the updated version. Below we address your concerns.
>
> *“There were no details on how to find supporting sentences of target facts that one wants to change”*
>
> In T-REx and zsRE, each supporting sentence is already linked to a fact so we did not need to find them. More specifically, each fact of the T-REx dataset has at least one supporting evidence and one template cloze question, and we use the supporting evidences and the cloze question to construct the training and test sets. For zsRE, we put the questions of a fact into training and test sets if the fact has multiple annotated questions, or keep it in the training set if it is the only question for the fact. We have added more details about the data creation process in Appendix A.
> In the future, if we were to build similar benchmarks using a completely unstructured corpus, we can use retrieval techniques to find supporting evidences for any fact within the corpus. This may not be perfect, but we can augment it with synthetic evidences constructed from various templates for the facts. Our benchmark and modification methods can work as long as we provide enough supporting evidences for the model to learn.
>
> ---
>
> *“Why do the authors start with pretrained + fine-tuned model? Isn't pretrained model trained on all sentences anyways? Do they mean separate fine-tuned data that contain factual statements? Why does one need this data? Isn't this against the spirit of pretrained models as knowledge bases?”*
>
> The pretrained model refers to off-the-shelf BERT-Base models pretrained on the Wikipedia Corpus and the Bookcorpus. For finetuning (FT), we finetune the model on all the supporting evidences of T-REx or zsRE,  which improves the accuracy of BERT-Base by e.g. 20% on T-REx. The improved accuracy after the finetuning process allows us to explore the more ideal setting where a language model has good performance on the questions and is ready to be deployed in practice. This also allows us to choose the facts to be modified from a larger set of facts, because we only modify facts that the model has already learned correctly.
>
> ---
>
> *“Results of vanilla pretrained models on unmodified and modified facts missing, i.e., PT alone. Also, PT + FT.”*
>
> We have added these results into Table 2 of the current version.
>
> ---
>
> *“The notation and acronyms are confusing. The authors use a lot of acronyms like PT, FT, FTA, FTM, RT which is unnecessary. This makes the results table unreadable without reading the paper.”*
>
> We have revised our experiment section to improve the readability.
>
> ---
>
> *“No citations on retrofitting literature which is very similar idea to this work (e.g., Faruqui et al. 2015).”*
>
> We have added discussions in the related works. While retrofitting does seem similar to our work in using facts/knowledge to finetune the models, the goals are different. We aim at changing the prediction of the models while preserving its performance on unmodified facts, whereas retrofitting aims to obtain better word representations using relations.
>
> ---
>
> *“How is PT+FT+FTM different from PT + D_F'? FTA is introduced but never used.”*
>
> Our previous version has results for PT+FT+FTA in Table 4 in the appendix (now still Table 4 but denoted as FT+FTA). We have also added PT+FTA and PT+FTM in Table 2 (now denoted as FTA and FTM).
> Our preliminary experiments with BERT-Base show that, in comparison with PT+FT+FTM, PT+D_F’ (drawing random batches from D_F’ and finetuning on D_F’ without any constraint) has a higher accuracy on the unmodified facts (around 50%), but its accuracy on the modified facts is also around 50%, lower than the current results on the modified facts. In our scenario, we want a higher accuracy on the modified facts.
> Also, as can be seen in Table 2 in the current version, if we use a mixture of the modified and unmodified facts in each minibatch from D_F’ and enforce the norm constraints, we can improve the accuracy on the unmodified facts in the FT+FTA setting (previously noted as PT+FT+FTA). That said, we would like to note that the result of FT+FTA (previously denoted as PT+FT+FTA) is still not as good as FT+FTM (previously denoted as PT+FT+FTM).

---

> > ### Author Response · Authors · 2020-11-19
> > **Rebuttal for Reviewer 3, part 2**
> >
> > *“The reviewer had a hard time following experiments with FAE section. What is the main takeaway from these experiments? The model has to be described in detail (perhaps with a figure).”*
> >
> > FAE has both implicit memory in its parameters and an explicit, symbolic memory module. It is therefore interesting to check their behavior in our new modification benchmark. We updated the paper to clarify the takeaways:
> > 1. To achieve high accuracy on the modified facts, only modifying the symbolic links of FaE is not enough; we also need to finetune its parameters.
> > 2. FaE has the best average accuracy ($\bar{\mathfrak{A}}$) so far after modification.
> > 3. Using norm constrained optimization, to achieve the same level of accuracy on the modified facts, the performance loss on the unmodified facts in FaE is not necessarily smaller than that in BERT models.
> > We will include the figure as soon as possible.
> >
> > *“Examples, Examples, Examples. Show some examples from each dataset.”*
> >
> > Great suggestion! We now explain the benchmark construction process with an example (Section 4.1) and include more examples in Appendix A.

---

### Decision · Program_Chairs · 2021-01-07
**Final Decision**

**Decision:**

Reject

**Comment:**

All reviewers noted the significance of the problem tackled by this paper and felt that it is going in the right direction. However, they also all noted that the paper was not finalized and polished well enough to be granted publication: details missing, typos, clarifications needed. The reviewers acknowledged the large amount of work that went into improving the paper during the discussion period. R1 even increased their score to reflect that.

Still the paper still needs some work to be accepted at ICLR. In particular, we encourage the authors to improve on 2 axes.
1. Clarifying motivations and contribution: it is still unclear if the main point of the paper is to propose new methods around FTM & constrained updates, etc. or around proposing a new benchmark for catastrophic forgetting, lifelong learning.
2. Reorganizing experimental section: the experimental section should be organized to support #1. Reviewers made a lot of suggestions, like moving Table 4 from the appendix, that should be further refined

We hope that this will allow to increase the clarity and impact of this research work.